# Early and Late Efficacy on Wound Healing of Silver Nanoparticle Gel in Males after Circumcision

**DOI:** 10.3390/jcm9061822

**Published:** 2020-06-11

**Authors:** Matteo Balzarro, Emanuele Rubilotta, Nicolò Trabacchin, Antonio Soldano, Clara Cerrato, Filippo Migliorini, Vito Mancini, Antonio Luigi Pastore, Antonio Carbone, Luigi Cormio, Giuseppe Carrieri, Alessandro Antonelli

**Affiliations:** 1Department of Urology, Azienda Ospedaliera Universitaria Integrata di Verona, 37126 Verona, Italy; emanuele.rubilotta@aovr.veneto.it (E.R.); nicolo.trabacchin@gmail.com (N.T.); soldanoantonio@libero.it (A.S.); clara.cerrato01@gmail.com (C.C.); filippo.migliorini@aovr.veneto.it (F.M.); alessandro_antonelli@me.com (A.A.); 2Department of Urology and Renal Transplantation, University of Foggia, 71122 Foggia, Italy; mancini.uro@gmail.com (V.M.); luigi.cormio@unifg.it (L.C.); giuseppe.carrieri@unifg.it (G.C.); 3Department of Medic-Surgical Sciences and Biotechnologies Urology Unit, Sapienza University of Rome, 00185 Latina, Italy; antopast@hotmail.com (A.L.P.); antonio.carbone@uniroma1.it (A.C.)

**Keywords:** circumcision, wound dressing, silver nanoparticles, AgNPs

## Abstract

We evaluate the early and late safety and efficacy of silver nanoparticle (AgNPs) in wound healing after circumcision. This multicenter prospective comparative non-randomized observational study compares wound dressing with AgNPs (group A) vs. gentamicin cream (group B). Follow-up included objective evaluation at 10 and 30 days by the Southampton Scoring System (SSS) and Stony Brook Scar Evaluation Scale (SBSES). We enrolled 392 males: 194 in group A, and 198 in group B. At 10 days follow-up, in group A, the SSS scale was grade 1 in 49.5% and grade 2 in the remaining; meanwhile, in group B, grade 1 was in 58%, grade 2 in 34.3%, and grade 4 in 7.6%. At 30 days follow-up, grade 1 healing was 97.4% and 98.4% in group A and B, respectively. At 10 days follow-up, the mean SBSES score was 3.58 and 3.69 in group A and B, respectively; while at 30 days follow-up, 4.81 and 4.76 in group A and B, respectively. Only in group B did 7.6% of males have antibiotic therapy due to pus discharge. No patients needed surgical wound revision. AgNPs led to a late but safer healing, they were non-inferior to the antibiotic cream wound dressing efficacy, and they avoided pus discharge and the need for oral antibiotics due to their polymer material.

## 1. Introduction

In an era when a change of mindset on antibiotic use is needed, the international society guidelines have adapted the recommendations on the main surgical procedures [1,2,3,4]. Unfortunately, poor indications have been issued on minor invasive operations, such as circumcision. Circumcision may be considered a clean surgery, with a low rate of wound infection, and not routinely requiring surgical antibiotic prophylaxis (SAP) [1,2]. However, it is common clinical practice to prescribe the use of an antibiotic cream for a few days after the procedure. Indeed, despite the presence of a wound dressing, the anatomical site of circumcision is close to areas known to be colonized by bacteria [5,6]. However, there are no specific data on the non-use of topical antimicrobials after this procedure. In case adequate skin preparation is difficult, then intravenous antibiotics are recommended at induction by guidelines on clean surgery [2,3,4]. However, there are also concerns about potential adverse events associated with SAP in adults and children undergoing urologic procedures, leading to uncertainty about its risk/benefit ratio [7,8]. In order to avoid the improper use of antibiotics, non-antibiotic creams have been investigated [9,10]. Among these topical products, silver nanoparticle (AgNP) hydrogels have been demonstrated as an ideal wound dressing due to proven antibacterial activity and tissue regeneration with no cytotoxicity [9]. The antibacterial mechanism of AgNPs mainly includes contact reactions and reactive oxygen-catalyzed reactions [10]. 

The aim of this study was to assess wound healing after circumcision, and comparing wound dressings with the use of a topical silver nanoparticle hydrogel cream (Peonil^®^) versus topical antibiotic cream.

## 2. Materials and Methods 

This was a multicenter prospective comparative non-randomized observational study. Three urological Department of Tertiary hospitals were recruited. The choice of wound dressing was based on institutional standard practice. Each urological center, after circumcision, performed its usual institutional topical therapy: two centers with AgNPs, and one with antibiotic cream (control group). 

Informed, written consent was obtained by all patients recruited. This study was performed according to the Declaration of Helsinki. The Ethics Committee did not consider it necessary to be involved, as the clinical routine of each individual center was not changed. However, this study was recorded within the department audit.

Data on males who underwent circumcision between January and September 2019 were collected prospectively on a dedicated database. Patients were allocated into two groups: group A was treated with a hydrogel cream containing AgNPs, titanium dioxide, hyaluronic acid, and aloe vera; group B was treated with a topical use of gentamicin cream. Data on patients age, co-morbidities, home therapies, and previous keloid formation were collected.

The penis was surgically disinfected with povidone-iodine. Local anesthesia was performed by a dorsal penile nerve block and a circumferential block with 1% lidocaine. Circumcisions were done with a conventional technique, while suturing was done by Vicryl rapide 3–0 single absorbable stitches spaced by 3–4 mm [11]. 

Surgical complications were recorded. Wound dressing was carried out three times a day, without the use of disinfectants, but with the only use of the topical cream, for 10 days. Sport, sexual activity, or any other strenuous activity were avoided by the patients for 4–6 weeks. Follow-ups included a clinical objective evaluation at 10 and 30 days by two healing validated scales filled out by different urologists from the surgeons who performed the circumcision.

The Southampton Scoring System (SSS) evaluated wound complications related to surgery [12]; good healing was considered at SSS a grade ≤ 1 (“normal healing or with mild bruising or erythema”). The Stony Brook Scar Evaluation Scale (SBSES) evaluated the quality of healing; it comprises 6 items to assess the short-term cosmetic outcome of the wound, with a score ranging from 0 (worst) to 5 (best) [13]. 

The exclusion criteria were: corticosteroids and/or other immunosuppressive therapies, uncontrolled diabetes, and conditions causing immunosuppression.

## 3. Statistical Analysis 

SPSS 19.0 was used for data analysis. Continuous variables were presented as mean and standard deviation. The categorical variables were presented as frequencies and percentages. These data were evaluated by the χ^2^ test, Fisher’s exact test, and Student’s *t*-test. *P* ≤ 0.05 was considered statistically significant.

## 4. Results

We enrolled 392 males with similar demographic characteristics, reported in Table 1: 194 patients in group A and 198 in group B. The mean age was 36.8 years, standard deviation ± 17.1. Table 2 reports the wound healing in the two groups, according to the SSS and SBSES. According to the SSS scale, 10 days after surgery, all patients treated with AgNPs had normal healing, although almost half of them showed mild bruising or erythema (SSS grade 1). In group B, we found a higher rate of normal healing (58%) and a significantly lower incidence of erythema, plus other signs of inflammation (SSS grade 2). The very relevant difference between the two groups was the presence of males with pus discharge (SSS grade IV) associated with a ≤ 1 cm diastasis only in group B (7.6%). These patients required additional oral antibiotic therapy with co-amoxiclav 1 g, three times per day, for 10 days. However, at the one-month follow-up, both groups showed a normal wound healing in almost all patients with no statistically significant difference. 

The cosmetic healing by the SBSES was similar in the two groups at 10- and 30-day follow-ups (Table 3). No major complications occurred, and no patients needed surgical wound revision.

## 5. Discussion

Although circumcision is one of the most performed surgical procedures in andrology, there are no specific instructions on antibiotic prophylaxis for this surgery [1,2,3,4]. Furthermore, antibiotic prophylaxis may cause side effects without decreasing the incidence of post-circumcision infection [14]. Hence, the few data reported do not support the use of antibiotic prophylaxis [14,15]. Therefore, in our study, no patients received antibiotic prophylaxis (oral or intra-venous).

Although circumcision is a clean surgery, there are some reasons which may lead to the use of topical antimicrobial agents after the surgery. A risk factor is represented by the wound site, which is close to areas colonized by bacteria. Moreover, the complication rate of circumcision ranges from 3.6% to 18%, of which 3.7–25% are wound infections [16,17]. Thus, after circumcision, an antibiotic cream is applied for 7–10 days. However, there are no data reporting a higher efficacy of antibiotic cream than antiseptic ointment. AgNPs are found in a non-antibiotic hydrogel cream with bactericidal activity, promoting wound healing [18]. The mechanism is related to tissue regeneration by avoiding over-infection. The barrier effect of topical therapy better protects the wound surface, with no local toxicity [9]. AgNPs consist of hydrogel and silver nanoparticles. Hydrogel has a role in the covering the wound as a temporary barrier, avoiding water loss and bacterial infection due to its antibacterial properties [9,18]. Silver nanoparticles significantly improve the antibacterial effect of hydrogel without cytotoxicity [19,20,21,22,23,24]. Moreover, hydrogel has the capability to absorb the excess wound exudate with moisturizing efficacy [9,18]. Therefore, we used AgNPs as a non-antibiotic cream for wound dressing after circumcision.

In our study, we assessed wound regeneration during the first month after surgery in males treated with topical AgNP hydrogel cream vs. antibiotic cream. At an early follow-up post circumcision, the wound dressing with AgNPs showed a slower rate of normal healing that was equal to the gentamycin group at the 30-day follow-up. Conversely, the patients dressed with gentamycin cream had a faster rate of healing, but only in this group were patients affected by pus discharge associated with a ≤ 1 cm diastasis, requiring additional oral antibiotic therapy. Therefore, dressing the surgical wound with AgNPs led to a late, but safer, healing. The lack of wound pus discharge in patients treated with AgNPs may be related to its structure. The polymer material has a network structure, which is made of hydrogel, guaranteeing a temporary barrier, which prevents and absorbs excessive water loss [9]. This effect, associated with the moisturizing property, and to the antibacterial effect, allows a better wound healing. Silver nanoparticles improved the hydrogel antibacterial efficacy and lowered the risk of wound infection. These microstructural characteristics may explain the lack of pus discharge in patients treated with AgNPs. Therefore, AgNPs have both antibacterial and wound healing properties, while antibiotic creams have only an antibacterial effect, without promoting wound healing.

Due to the lack of wound dehiscence and pus discharge in males treated with AgNPs, oral antibiotics were not administered to these patients. Consequently, the potential side effects, toxicity, and drug resistance related to the use of antibiotics was avoided.

The overall cosmetic wound healing was not influenced by the kind of wound dressing. At the early follow-up, the score of aesthetic healing was similar in the two groups, although the rate of normal healing was slower in males treated with AgNPs. This finding might be explained by the slower rate of normal healing in the AgNP group balanced by the presence of patients with dehiscence and pus discharge in the gentamycin group only. 

Our study has some limitations. A first limit is the lack of randomization. Another limit is the absence of a control group without any kind of cream or ointment on the wound dressing. However, our data report the results of surgical wound healing medicated with AgNPs in a large sample of patients, who would otherwise be treated with topical antibiotic therapy.

## 6. Conclusions

Our data showed that wound dressing with AgNPs was a safe and effective treatment. AgNPs were non-inferior to the antibiotic cream wound dressing. Males managed with AgNPs had no wound pus discharge and no need for oral antibiotics. Therefore, wound dressing with AgNPs is an effective alternative to the use of topical antibiotics.

## Figures and Tables

**Table 1 jcm-09-01822-t001:** Demographics characteristics of group A and B.

	Group A, *n* (%)	Group B, *n* (%)
Population	194	198
Mean age	36.2	37.4
Ethnicity	Caucasian 194 (100%)	Caucasian 198 (100%)
Indication for surgery		
Idiopathic phimosis	188 (96.9%)	189 (95.4%)
Lichen Sclerosus phimosis	8 (3.1%)	9 (4.5%)
Cancer phimosis	0 (0%)	0 (0%)
Co-morbidities		
No co-morbidities	161 (83%)	165 (83.3%)
Controlled diabetes	11 (5.7%)	12 (6.1%)
Hypertension	12 (6.2%)	13 (6.6%)
Vasculopathy	6 (3.1%)	5 (2.5%)
Metabolic disorders	16 (8.2%)	14 (7.1%)
Therapies		
Oral diabetic therapy/insulin	11 (5.7%)	12 (6.1%)
Antihypertensive therapy	10 (5.6%)	10 (5.1%)
Antiplatelet/anticoagulant	13 (6.7%)	15 (7.6%)
Metabolic therapies	11 (5.7%)	12 (6.1%)
Alpha adrenergic therapy	9 (4.6%)	7 (3.5%)
Previous urological surgeries	7 (3.6%)	8 (4%)

**Table 2 jcm-09-01822-t002:** Wound healing in the two groups analyzed by the Southampton Scoring System and Stony Brook Scar Evaluation Scale at 10- and 30-day follow-ups.

	10 Days f-Up			30 Days f-Up		
	Group A	Group B	P	Group A	Group B	P
Grade 0 Normal healing	-	-		97.4% (189/194)	98.4% (195/198)	0.45
Grade I Normal healing with mild bruising or erythema	49.5% (96/194)	58.0% (115/198)	0.88	2.6% (5/194)	1.6% (3/198)	0.45
Grade II Erythema plus other signs of inflammation	50.5% (98/194)	34.3% (68/198)	0.001	-	-	
Grade III Clear or haemoserous discharge	-	-		-	-	
Grade IV Pus	-	7.6% (15/198)	-	-	-	-
Grade V Deep, or severe, wound infection with or without tissue breakdown; hematoma requiring aspiration.		-		-	-	

* Fisher exact test.

**Table 3 jcm-09-01822-t003:** Circumcision cosmetic healing evaluated by the Stony Brook Scar Evaluation Scale.

	10 Days f-Up			30 Days f-Up		
Stony Brook Scar Evaluation Scale	*Group A*	*Group B*	P	*Group A*	*Group B*	P
Mean score	3.58	3.69	0.064 *	4.81	4.76	1 *
SD score	± 1.1	± 1.2		± 0.4	± 0.3	

* T student test.

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
