# Peer review of "Early and Late Efficacy on Wound Healing of Silver Nanoparticle Gel in Males after Circumcision"

_jcm, 2020, doi:10.3390/jcm9061822_

Round 1
Reviewer 1 Report
Well written article, and interesting findings regarding use of AgNP. This might be something we should look into further. A few things to think about: adding a table of demographics between the two groups - are they similar in comorbidities? There are certainly other comorbidities that can affect wound healing. If the patients in each group are not similar then the results cannot be compared or generalizable. Also, what was the decision on which patients received AgNP vs antibiotic cream? Was it institutional, or surgeon preference?
Author Response
Reviewer 1: Well written article, and interesting findings regarding use of AgNP. This might be something we should look into further. A few things to think about:
Authors’ answers to reviewer 1:
Thank you for your positive opinion on our manuscript. Moreover, we want to thank you to give us the opportunity to improve our manuscript with your comments.
Here are your comments and our answers:
Adding a table of demographics between the two groups - are they similar in comorbidities? There are certainly other comorbidities that can affect wound healing. If the patients in each group are not similar then the results cannot be compared or generalizable.
We confirm that the demographics characteristics of the two groups were similar. As suggested by the reviewer we added in the text table 1 describing the patients characteristics for Group A and B.
- Page 5, line 118 we added the table of demographics between the two groups. This is now table 1. In the text we had to present table 1 in the Results section at page 4, line 101.
- Due to the adding of table 1 we had to rename the others two tables: table 1 is now table 2 (page 6, line 122), and table 2 is now table 3 (page 6, line 126).
Also, what was the decision on which patients received AgNP vs antibiotic cream? Was it institutional, or surgeon preference?
The decision of the wound dressing was based on the Institutional practice. We better explained this important data in the manuscript in the Material and Methods section (page 3, line 64-66).
We also corrected a misprint in page 3, line 60 we cancelled “(AgNPs)”: it was present in page 2, line 55 .

Reviewer 2 Report
The authors have conducted a non-randomised study to explore outcome and efficacy of 2 different creams in the post-operative management following circumcision in adult males. Their main research question was to assess if the use of a non antibiotic cream is associated with an increased number of infection and complications.
The topic is interesting and the intention is to reduce the use of antibiotic, which is commendable.
The methods are sound (although they have not randomised the participants) and the number of patients enrolled is fine (about 200 per arm).
The authors stated that both groups had similar demographic characteristics but they do not report it. I would suggest to have a table reporting at least some details such age, ethnicity, indication for surgery, co-morbidities, etc.
The methods used to assess outcome are fine.
It is quite surprising that at the 30 days follow-up no one has scored grade 0; how can the authors explain that?
The number of short term infections and long-term complications are low (15 and 8 individuals, respectively): it would be interesting if the authors could speculate on why they have occurred (activity, compliance with cream application, underlying condition).
The statistical analysis does not seems correct and needs to be revised.
Author Response
Reviewer 2: The topic is interesting and the intention is to reduce the use of antibiotic, which is commendable. The methods are sound (although they have not randomised the participants) and the number of patients enrolled is fine (about 200 per arm). The methods used to assess outcome are fine.
Authors’ answers to reviewer 2:
Thank you for your positive opinion on our manuscript. Moreover, we want to thank you to give us the opportunity to improve our manuscript with your comments.
Here are your questions and our answers:
The authors stated that both groups had similar demographic characteristics but they do not report it. I would suggest to have a table reporting at least some details such age, ethnicity, indication for surgery, co-morbidities, etc.
We added the demographics characteristics in a new table: Table 1.
- Page 5, line 118 we added the table of demographics between the two groups. This is now table 1. In the text we had to present table 1 in the Results section at page 4, line 101.
- Due to the adding of table 1 we had to rename the others two tables: table 1 is now table 2 (page 6, line 122), and table 2 is now table 3 (page 6, line 126).
It is quite surprising that at the 30 days follow-up no one has scored grade 0; how can the authors explain that?
In the Results section we reported that at 30 days follow-up almost of patients had a normal wound healing. However, we found that in the table reporting these data (now it is table 2) the reports at 30 days were in the wrong line. Thus, we corrected table 2: scores 0 was reached by 97.4% of males in group A, and by 98.4% of group B.
It is quite surprising that at the 30 days follow-up no one has scored grade 0; how can the authors explain that?
As reported above we found a mistake on table 2, thus due to the corrections there are only short-term infection, and no long-term complications. We thing that
The number of short-term infections and long-term complications are low (15 and 8 individuals, respectively): it would be interesting if the authors could speculate on why they have occurred (activity, compliance with cream application, underlying condition).
In the group that used AgNPs we had no pus discharge due to AgNPs capability to absorb the excess wound exudate with moisturizing efficacy reduced the bacterial growth. While in the males using gentamycin cream the not adsorbed exudate may have been a good culture for bacteria growth. Moreover, the antibiotic cream might have selected non responder bacteria. We preferred to do not report this hypothesis on the manuscript due to the lack of evidences: we did not perform the culture of wound discharge/pus. Furthermore, males referred high compliance to the wound dressing but we cannot exclude that some activities, or some unknown underlying conditions, may have occurred.
The statistical analysis does not seem correct and needs to be revised.
In the Material and Methods section, subsection Statistical analysis we Page 4, line 99-100 we added the T student test which was used to analyze continuous variables. We also corrected a typing error in table 3 of the statistical used test (page 7, line 140).
We also corrected a misprint in page 3, line 60 we cancelled “(AgNPs)”: it was present in page 2, line 55.
Round 2
Reviewer 2 Report
I am happy with the changes and corrections. Thank you to the authors for their work.